# MUX-PLMs: Data Multiplexing for High-throughput Language Models

**Vishvak Murahari**[1]      **Ameet Deshpande**[1]      **Carlos E. Jimenez**[1]
**Izhak Shafran**[2]      **Mingqiu Wang**[2]      **Yuan Cao**[2]      **Karthik Narasimhan**[1]

[1]Princeton University      [2] Google Brain
murahari@cs.princeton.edu

## Abstract

The widespread adoption of large language models such as ChatGPT and Bard has led to unprecedented demand for these technologies. The burgeoning cost of inference for ever-increasing model sizes coupled with hardware shortages has limited affordable access and poses a pressing need for efficiency approaches geared towards high throughput and performance. Multi-input multi-output (MIMO) algorithms such as data multiplexing, offer a promising solution with a many-fold increase in throughput by performing inference for multiple inputs at the cost of a single input. Yet these approaches are not currently performant enough to be deployed in modern systems. We change that by developing MUX-PLMs, a class of deployable high throughput pre-trained language models (PLMs) trained with data multiplexing, that can be fine-tuned on any downstream task. Our novel multiplexing and demultiplexing modules proficiently entangle and disentangle inputs, and enable high-performance high throughput MUX-PLMs that are competitive with vanilla PLMs while achieving 2x/5x inference speedup with only a $1 - 4\%$ performance drop on a broad suite of tasks.

## 1 Introduction

Language models like ChatGPT (OpenAI, 2023), PaLM (Chowdhery et al., 2022), T5 (Raffel et al., 2020), and CM3 (Aghajanyan et al., 2022), have seen unprecedented adoption in diverse sectors ranging from education and healthcare to manufacturing and marketing. The proficiency of these tools has led to unprecedented demand for these models, with users facing frequent outages and capacity limits. Additionally, ever-increasing model sizes and hardware shortages have constrained models' ability to handle a very high load of requests, thus limiting large-scale affordable access to these models. These trends bring into focus the need for high-throughput, high-performance, efficient, and environmentally responsible models that can be deployed at scale to meet the quickly growing demand.

Multi-input Multi-output architectures (MIMO) (Havasi et al., 2021; Ramé et al., 2021; Murahari et al., 2022) are a promising hardware-agnostic and architecture-agnostic paradigm that perform inference for multiple inputs *simultaneously* at the cost of a single input. This efficiency paradigm is natively geared towards yielding high-throughput models, in addition to being complementary in approach and motivation to current efficiency methods such as pruning, quantization, and distillation. Interestingly, MIMO approaches are partly inspired by the human brain's extraordinary ability to process multiple inputs and propagate information at a high bandwidth with a few neural codes (Blumhagen et al., 2011; Akam and Kullmann, 2014; Pirschel and Kretzberg, 2016; Hong et al., 2016; Friedrich et al., 2004).

Murahari et al. (2022) introduced data multiplexing, a MIMO technique that can enable a many-fold increase in throughput. The method compresses $N$ different instances into a single "multiplexed" hidden representation before decompressing it into $N$ independent predictions. While they show the plausibility of MIMO training, their method leads to a significant drop in performance ($20 - 30\%$ points) compared to state-of-the-art models.

In this work, we introduce MUX-PLMs, a class of high-throughput pre-trained language models trained in a MIMO fashion with data multiplexing to process multiple inputs (2-10) simultaneously with a forward pass over a single instance. MUX-PLMs offer up to $400\%$ improvement in throughput over baseline pre-trained models while only being $\sim 4$ points and $\sim 2$ points worse than baseline pre-trained language models for text classification and token classification tasks, respectively. MUX-PLMs, like other pre-trained language models, provide general model initialization that can be fine-tuned for *any* downstream

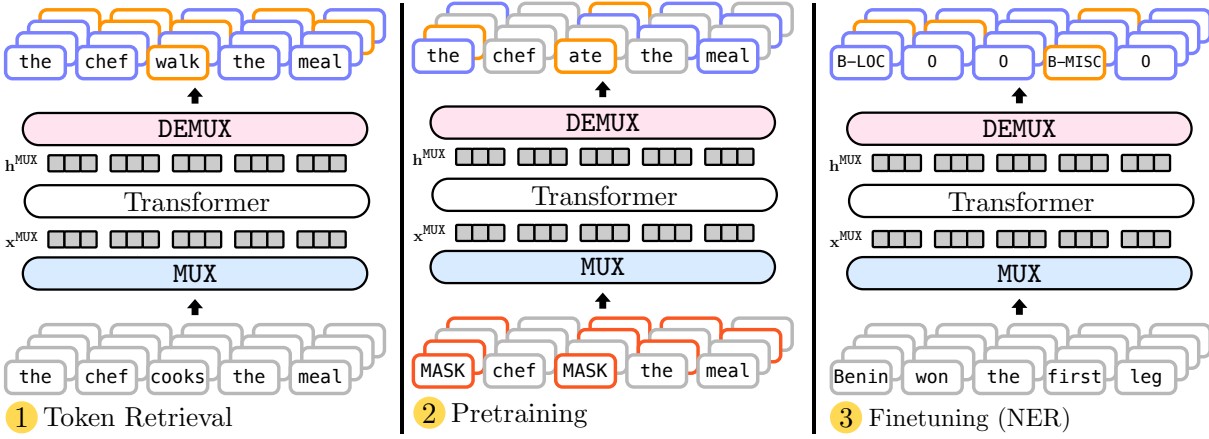

Figure 1: Illustrating the training process for MUX-PLMs. MUX-PLMs are first primed for MIMO style training with a token-retrieval auto-encoding task, where the model is trained to output the tokens in the $N$ inputs. MUX-PLMs are then pre-trained by adapting standard pre-training objectives (BERT in this example), to MIMO style training with data multiplexing. The resulting MUX-BERT model, similar to standard PLMs, provides a general model initialization that can be fine-tuned on any downstream task (NER in this example). Output predictions are shown above the system head with highlighted predictions contributing to the gradient update; violet indicates a correct prediction while orange indicates an incorrect prediction. Red highlighted tokens in the input indicate a position that has been masked.

task. We demonstrate the effectiveness and generality of our MUX-PLMs class of pre-trained models by training MUX-BERT and MUX-ELECTRA models, which are trained with pre-trained objectives adapted from BERT (Devlin et al., 2019) and ELECTRA (Clark et al., 2020) respectively, although in a MIMO fashion with data multiplexing.

Our work is the first to introduce MIMO architectures to PLMs. To enable this, we first develop a new demultiplexing module, RSA-demux (Figure 2), that randomly initializes and learns private key vectors to recover the multiple outputs from a multiplexed representation. Secondly, we introduce a new *Contextual Multiplexer* module (Figure 3) that uses a cross-instance attention-based mechanism to aggregate context across the set of multiplexed instances, which seems to be particularly effective for token-level tasks. Thirdly, our three-stage training algorithm (Figure 1) enables stable and efficient training of MUX-PLMs.

Importantly, MUX-PLMs are complementary to existing state-of-the-art model compression techniques. We hope our work validates MIMO architectures as a promising complementary direction to existing efficiency techniques. Consequently, we hope future research develops MIMO architectures in tandem with other efficiency approaches, leveraging the best of both paradigms. We will publicly release our models and code to promote open-source research on the next generation of MIMO

architectures for large language models.

## 2 Related Work

**Efficient Inference with Transformers** Recent methods in NLP rely heavily on transfer learning through Transformer-based (Vaswani et al., 2017) language models trained on large text corpora using self-supervised objectives, such as autoregressive (Radford and Narasimhan, 2018) or masked language modeling (Devlin et al., 2019). Prior analysis on pre-training language models has observed power-law scaling of model performance with respect to model size (Kaplan et al., 2020), leading the community to develop ever-larger language models. It is also generally recognized that pre-trained language models are significantly over-parameterized; effectively learning a *subnetwork* that utilizes only a relatively small number of their total parameters (Voita et al., 2019; Michel et al., 2019; Gordon et al., 2020; Prasanna et al., 2020).

The ubiquity of pre-trained language models, their growing size, and over-parameterization has inspired extensive research on improving inference efficiency. This includes methods such as structured pruning (Liu et al., 2019; Wang et al., 2020; Lagunas et al., 2021; Xia et al., 2022; Yang et al., 2022), knowledge distillation (Hinton et al., 2015; Sanh et al., 2019; Sun et al., 2020; Jiao et al., 2020; Yin et al., 2021), quantization (Zafrir et al., 2019; Shen et al., 2020), and data multiplexing (Mura-

hari et al., 2022). These approaches assume that PLMs are highly over-parametrized and attempt to approximate a large function by learning a smaller, compressed, version of the original model. Past work has also focused on unstructured pruning for both task finetuning (Chen et al., 2020; Sanh et al., 2020) and pre-trained (Zafrir et al., 2021; Jiang et al., 2022) language model settings, but don't increase model throughput due to hardware limits.

**Multi-input Multi-output Models** While pruning, quantization, and distillation seek to reduce overparameterization by reducing models' representational capacity, other lines of work seek to exploit overparameterization in other ways. Multi-input Multi-output (MIMO) architectures (Havasi et al., 2021; Ramé et al., 2021; Murahari et al., 2022) train models using mixed-instance representations, i.e. Zhang et al. (2018), in order to obtain predictions for multiple instances simultaneously. Unlike efficiency methods, Havasi et al. (2021) and Ramé et al. (2021) try to obtain better performance by inducing multiple subnetworks in a single convolutional model to perform "ensembling for free" during inference. Data multiplexing, introduced in DataMUX (Murahari et al., 2022), aims to improve model efficiency by training Transformer models with mixed-instance representations to perform simultaneous inference for language tasks, thereby improving inference throughput many-fold. Currently, MIMO architectures have only been used in a limited setting, achieving middling performance. Our work training PLMs with a potent MIMO architecture, data multiplexing, dramatically improves inference throughput while preserving high accuracy for downstream tasks.

## 3 Methodology

We briefly introduce readers to the data multiplexing MIMO architecture (Murahari et al., 2022), which we denote T-MUX. We then detail our novel approach to train MUX-PLMs to yield high-throughput and performant language models.

### 3.1 T-MUX: Data multiplexing with Transformer

Data multiplexing allows parallel processing of multiple sequences with a single forward or backward pass through the model ($M$) and requires two crucial components, multiplexer, and demultiplexer.

**Multiplexer** The multiplexer module (MUX) combines an ordered set of multiple inputs – $X^{1:N} = (\mathbf{x}^1, \ldots, \mathbf{x}^N)$ into a single superimposed representation ($\mathbf{x}^{\text{MUX}}$). If $\mathbf{x}^i \in \mathbb{R}^d$, the multiplexer is a transformation (MUX: $\mathbb{R}^{N \times d} \rightarrow \mathbb{R}^d$) such that $\mathbf{x}^{\text{MUX}} = \text{MUX}\left(X^{1:N}\right)$.

To ensure MUX is an order-preserving transformation, T-MUX samples a vector ($\mathbf{v}^i \in \mathbb{R}^d$) from a standard multivariate Gaussian and applies the Hadamard product (element-wise multiplication) with the corresponding input representation ($\mathbf{x}^i$) before summing up vectors for all positions.

$$\mathbf{x}^{\text{MUX}} = \text{MUX}\left(X^{1:N}\right) = \frac{1}{N} \sum_{i=1}^{N} \mathbf{x}^i \odot \mathbf{v}^i \quad (1)$$
$$\mathbf{v}^i \in \mathbb{R}^d \sim \mathcal{N}\left(\mathbf{0}, \mathbf{I}\right)$$

The model processes the multiplexed representation and emits a multiplexed hidden state – $\mathbf{h}^{\text{MUX}} = M\left(\mathbf{x}^{\text{MUX}}\right)$. To multiplex Transformers' sequenced inputs $\left(\mathbf{x}^i = \left(\mathbf{x}_1^i, \ldots, \mathbf{x}_L^i\right)\right)$ of length $L$, T-MUX applies the same $\mathbf{v}^i$ to all $L$ positions of instance $i$.

$$\mathbf{x}^{\text{MUX}} = \text{MUX}\left(X^{1:N}\right) =$$
$$\left(\frac{1}{N} \sum_{i=1}^{N} \mathbf{x}_1^i \odot \mathbf{v}^i, \ldots, \frac{1}{N} \sum_{i=1}^{N} \mathbf{x}_L^i \odot \mathbf{v}^i\right) \quad (2)$$

**Demultiplexer** A prediction needs to be made for each instance in $X^{1:N}$, and T-MUX's demultiplexer module (DeMUX) achieves this by separating the superimposed output ($\mathbf{h}^{\text{MUX}}$) into $N$ output representations corresponding to the input ($\mathbf{h}^1, \ldots, \mathbf{h}^N$).

$$\mathbf{h}^i = \text{DeMUX}\left(\mathbf{h}^{\text{MUX}}, \mathbf{p}^i\right)$$
$$\mathbf{h}_j^i = \text{DeMUX}\left(\mathbf{h}^{\text{MUX}}_j, \mathbf{p}^i\right) \quad (3)$$

The vector $\mathbf{p}^i \in \mathbb{R}^d$ is dynamically generated for each instance ($i$) with the help of a prefix that is added to the input and re-used for all positions in the instance. They add a *prefix$_i$* to $\mathbf{x}^i$, represented by the following pattern, where $\epsilon^i$ is a special token, and $\mathbf{p}^i$ is set to be the output corresponding to token $i$ in the prefix.

$$prefix^1 = [\epsilon^1, \epsilon^{\text{pad}}, \ldots, \epsilon^{\text{pad}}]$$
$$prefix^2 = [\epsilon^{\text{pad}}, \epsilon^2, \epsilon^{\text{pad}}, \ldots, \epsilon^{\text{pad}}]$$
$$\cdots$$
$$prefix^N = [\epsilon^{\text{pad}}, \ldots, \epsilon^{\text{pad}}, \epsilon^N]$$

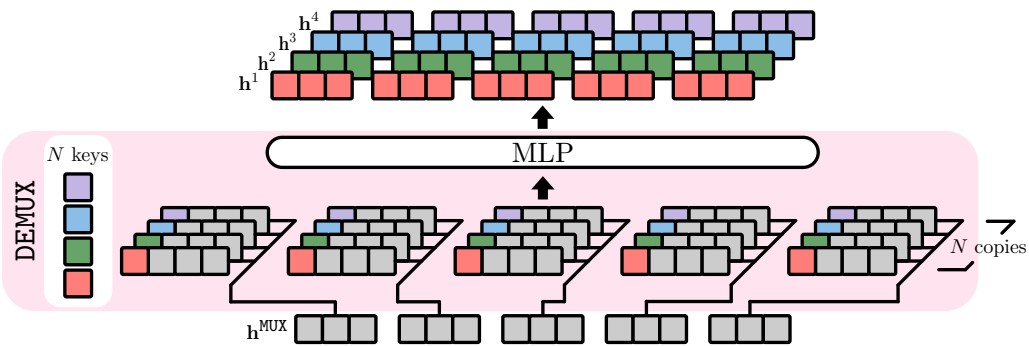

Figure 2: Illustrating our novel RSA-inspired demultiplexing module. The module is initialized with N key vectors which are used to demultiplex the transformed multiplexed representations ($h^{MUX}$). The keys are concatenated with $h^{MUX}$ and are processed with an MLP to generate the demultiplexed output representations ($h_1 \cdots h_4$).

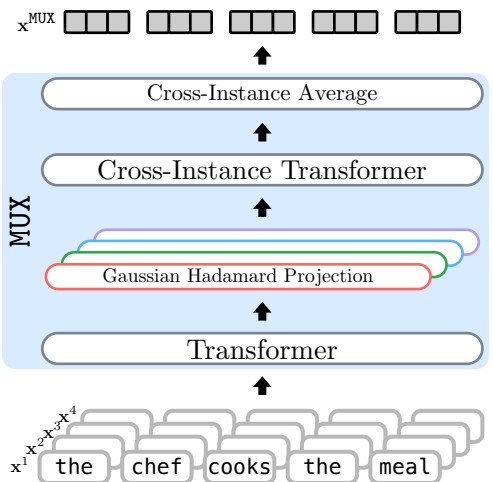

Figure 3: Illustrating our attention-based multiplexing module. The module generates contextual representations for instances $x_1 \cdots x_4$ with a Transformer layer and then applies a hadamard product between the contextual representations and the corresponding multivariate gaussian to generate instance-conditioned representations. The final multiplexed representations are generated by first applying another Transformer layer, which attends across the instances for all the positions in the sequence, and then averaging across the instances.

## 3.2 MUX-PLMs: Data multiplexing for high-throughput language models

We propose MUX-PLMs, a class of high-throughput pre-trained Transformer-based language models trained in a MIMO fashion with data multiplexing. To demonstrate the viability and the generality of this class of models, we pre-train Transformer models with objectives based on BERT and ELECTRA, to get MUX-BERT and MUX-ELECTRA respectively. MUX-PLMs are trained with our three stage training algorithm (Figure 1). Firstly, MUX-PLMs are trained with the token retrieval task in T-MUX, which is an auto-

encoding setup to decode all the tokens in the multiplexed input. This simple auto-encoding task is critical to prime the model for MIMO-style data multiplexing. The MUX-PLMs are then pre-trained with standard pre-training objectives but adapted to MIMO-fashioned training with data multiplexing. MUX-PLMs show significant throughput improvement over standard pre-trained LMs while matching their downstream task accuracies. Finally, MUX-PLMs, like other pre-trained language models, provide general model initialization that can be fine-tuned for *any* downstream task.

**Contextual multiplexer** T-MUX's multiplexer multiplexes tokens independent of 1) tokens in the same position in other instances and 2) other tokens in the instance, which could lead to suboptimal representations. We, therefore, explore a contextual multiplexing scheme that aggregates context both from tokens in the same instance and tokens in the same position of other instances (Figure 3). We first use a single transformer layer TRANS$_{\text{ctx}}$ to get contextual representations $\mathbf{h}_{\text{ctx}}^i = \text{TRANS}_{\text{ctx}}\left(\mathbf{x}_1^i, \ldots, \mathbf{x}_L^i\right)$ of length $L$. We apply a hadamard product with a multivariate gaussian $\mathbf{v}^i$ to all $L$ positions.

$$\mathbf{g}_{\text{ctx}}^i = \mathbf{h}_{\text{ctx}}^i \odot \mathbf{v}^i \qquad (4)$$

We generate multiplexed representations, $\mathbf{x}^{\text{MUX}}$, by applying another transformer layer TRANS$_{\text{inst}}$ across encoded representations from $N$ instances at each position from 1 to $L$. This is done by transposing $\mathbf{g}_{\text{ctx}}$ and applying TRANS$_{\text{inst}}$.

$$\mathbf{x}^{\text{MUX}} = \text{TRANS}_{\text{inst}}\left(\mathbf{g}_{\text{ctx}}^\top\right) \qquad (5)$$

**RSA demultiplexer** The *demultiplexer* in T-MUX requires a prefix whose length scales lin-

| Model | N | GLUE | | Token | | ↗ |
|---|---|---|---|---|---|---|
| | | Mean (std) | Max | Mean (std) | Max | |
| **BERT** | 1 | **85.4 (0.0)** | **85.4** | **95.8 (0.0)** | **95.8** | 1.0× |
| **ELECTRA** | | 82.1 (0.0) | 82.1 | 95.3 (0.0) | 95.3 | 1.0× |
| **T-MUX** | 2 | 60.4 (0.6) | 61.8 | 81.4 (0.1) | 81.5 | 1.9× |
| **MUX-BERT**‡ | | **82.5 (0.6)** | **83.4** | **95.2 (0.1)** | **95.4** | **2.0×** |
| **MUX-ELEC**‡ | | **82.5 (0.4)** | 83.1 | 95.0 (0.0) | 95.1 | **2.0×** |
| **T-MUX** | 5 | 59.7 (0.6) | 60.6 | 81.3 (0.2) | 81.5 | 4.4× |
| **MUX-BERT**‡ | | **80.3 (0.4)** | **80.9** | **93.6 (0.1)** | **93.6** | **4.9×** |
| **MUX-ELEC**‡ | | 79.8 (0.6) | 80.5 | 93.4 (0.0) | 93.5 | **4.9×** |
| **T-MUX** | 10 | 58.1 (0.5) | 59.1 | 79.7 (0.2) | 80.0 | 8.4× |
| **MUX-BERT**‡ | | 77.8 (0.6) | 78.8 | 91.6 (0.1) | **91.8** | **9.8×** |
| **MUX-ELEC**‡ | | **78.2 (0.6)** | **79.0** | **91.7 (0.1)** | **91.8** | 9.7× |

Table 1: Average GLUE and token-level classification scores for the BASE (L=12, H=768) configuration, across ELECTRA, BERT, and MUX-PLMs for $N \in \{1, 2, 5, 10\}$. ‡ indicates our models and ↗ indicates throughput increase w.r.t. to a vanilla BERT$_{\text{BASE}}$ model. All models are evaluated on 5 seeds with mean and max scores reported.

early with the number of instances ($N$), thus reducing the effective context length during pre-training, which degrades performance (Ainslie et al., 2020; Zaheer et al., 2020; Beltagy et al., 2020). Furthermore, it decreases throughput during inference for large $N$ because the model must process an extra prefix of length $N$ for each of the $N$ instances. To address these issues, we draw inspiration from the RSA cryptosystem (Rivest et al., 1978) to randomly initialize and learn $N$ (private) key vectors $(\mathbf{k}_1, \ldots, \mathbf{k}_N, \mathbf{k}_i \in \mathbb{R}^d)$ which are keys that can be used to demultiplex the output representation (Figure 2).

$$\begin{aligned} \mathbf{h}^i &= \text{DeMUX}\left(\mathbf{h}^{\text{MUX}}, \mathbf{k}^i\right) \\ \mathbf{h}^i_j &= \text{DeMUX}\left(\mathbf{h}^{\text{MUX}}_j, \mathbf{k}^i\right) \end{aligned} \quad (6)$$

Akin to RSA, $\mathbf{v_i}$ and $\mathbf{k_i}$ can be treated as the keys for multiplexing (encryption) and demultiplexing (decryption) while ensuring that, unlike T-MUX, the input sequence length does not change and thereby leading to an improvement in throughput. Importantly, this architecture ensures that the keys better transfer across the different stages of training as they are no longer conditioned on the input instances.

## 4 Experimental Setup

**Datasets** We pre-train all models on Wikipedia (Foundation) and Bookscorpus (Zhu et al., 2015). We evaluate on all datasets from the GLUE benchmark (Wang et al., 2018), which

are CoLA (Warstadt et al., 2019), SST-2 (Socher et al., 2013), MRPC (Dolan and Brockett, 2005), QQP (qqp), STS-B (Cer et al., 2017), MNLI (Williams et al., 2018), QNLI (Wang et al., 2018), RTE (Wang et al., 2018), and WNLI (Levesque et al., 2012). We also evaluate on token classification tasks such as named entity recognition (Sang and Meulder, 2003) and POS tagging (Grünewald et al., 2021). We don't report average over WNLI and CoLA as these are the two smallest tasks in GLUE and we observe high variance across different seeds. All numbers are reported on the dev split. We report scores for all tasks in Appendix E.

**Models** We experiment with ELECTRA and BERT pre-training objectives and present the pre-trained multiplexed models **MUX-BERT** and **MUX-ELECTRA** for $N = 2, 5$ and 10. To simplify training, we use a random generator to train MUX-ELECTRA models, presented as an ablation in Clark et al. (2020), instead of using a smaller masked LM. Except where otherwise noted, we do not use the contextual MUX module, but instead, use the RSA demultiplexing module. Refer to Appendix B and C for implementation details.

**Baselines** We run experiments to compare our models against T-MUX, from Murahari et al. (2022) and baseline PLMs - ELECTRA and BERT, across three different model configurations (SMALL, BASE, and LARGE). We also provide a comparison to results reported in recent PLM pruning and distillation papers in Table 2.

**Multi-run evaluation** We evaluate all models across 5 random seeds to reduce variance for smaller datasets which is caused by the randomized order in which we multiplex instances in the batch. We report both the average and maximum scores across seeds in Table 1 to understand the importance of ordering the multiplexed instances and report average across seeds for all other results.

## 5 Results

### 5.1 MUX-PLMs outperform PLMs and T-MUX

Table 1 shows that both **MUX-BERT and MUX-ELECTRA outperform T-MUX at all levels of multiplexing** ($N$), with improvements between 12 and 20 points on GLUE and token-classification tasks respectively. Furthermore, MUX-PLMs' efficient RSA-inspired demultiplexing method allows

| Model | ↗ | QNLI | QQP | SST2 |
|---|---|---|---|---|
| BERT | 1.0× | 90.5 | 91.2 | 91.7 |
| MUX-BERT (N=2) | 2.0× | 88.2 | 90.4 | 90.6 |
| MUX-BERT (N=5) | 4.9× | 85.6 | 88.8 | 86.9 |
| *Use additional unlabelled or task-specific data* | | | | |
| DistilBERT$_6$ | 2.0× | 89.2 | 88.5 | 91.3 |
| Block Pruning | 2.7× | 89.7 | - | 91.2 |
| Prune OFA | 1.0× | 90.3 | 91.2 | 91.5 |
| **Hybrid Approaches** | | | | |
| MUX-BERT (N=2) + CoFi | 4.0× | 88.4 | 90.4 | 90.0 |
| TinyBERT$_6$ | 2.0× | 91.1 | 91.1 | 93.0 |
| CoFi | 2.7× | 91.3 | - | 93.0 |
| AutoTinyBERT | 4.3× | 89.7 | 89.9 | 91.4 |
| MobileBERT | 2.3× | 91.0 | - | 92.1 |

Table 2: MUX-PLMs (variants are underlined) are complementary to existing efficiency methods, while being competitive standalone. Contrary to existing methods, MUX-PLMs *do not use additional unlabelled and task-specific data* and can be easily fine-tuned for *any* downstream task without architectural modifications. The inference speedups (↗) are reported against BERT$_{BASE}$.

| Config | Model | GLUE | Token | ↗ |
|---|---|---|---|---|
| SMALL | BERT | 80.6 | 94.0 | 5.9× |
| | T-MUX | 59.5 | 81.8 | 8.7× |
| | MUX-BERT‡ | 79.0 | 93.3 | 11.5× |
| BASE | BERT | 85.4 | 95.8 | 1.0× |
| | T-MUX | 60.4 | 81.4 | 1.9× |
| | MUX-BERT‡ | 82.5 | 95.2 | 2.0× |
| LARGE | BERT | 85.8 | 95.6 | 0.3× |
| | T-MUX | 61.7 | 80.9 | 0.6× |
| | MUX-BERT‡ | 84.1 | 95.2 | 0.6× |

Table 3: Changing the model size for MUX-BERT ($N = 2$) models. Across different model sizes, MUX-BERT outperforms T-MUX and achieve higher throughput (indicated under ↗ column). ‡ = our models.

it to achieve faster throughput than T-MUX, increasing it by over $16\%$ for $N = 10$.

Moreover, **MUX-PLMs provide a significant boost in throughput ($N$ times faster) when compared to PLMs, without a significant loss in performance.** For example, MUX-ELECTRA ($N = 2$) is 0.4 points better and only 0.3 points worse than ELECTRA for GLUE and TOKEN tasks respectively, while being $2\times$ faster. Similarly, MUX-BERT ($N = 2$) is within 3 and 0.6 points of BERT for GLUE and TOKEN tasks respectively, while being significantly faster. We also observe that as $N$ increases, MUX-PLMs' throughput is significantly better, though performance compared to PLMs can degrade. This is because a large $N$ implies that MUX-PLMs must parallelly process more instances, thus having to share network parameters and activations with a larger number of instances, thus improving throughput and degrading performance. For example, the gap between ELECTRA and MUX-ELECTRA on TOKEN for $N = 2$ is 0.2 points and increases to 3.5 points for $N = 10$, which shows that $N$ serves as a parameter to control the performance-throughput trade-off. We explore this further in Section 5.3 and Figure 4.

## 5.2 Comparing MUX-PLMs with other model compression methods

We compare our MUX-PLM models with other efficient learning methods, such as pruning and distillation, in Table 2. Contrary to other methods, our *vanilla* MUX-PLMs achieve competitive performance and significant throughput improvement

*without* additional unlabeled and task-specific data, and can be easily fine-tuned on *any* downstream task without any architectural modifications (unlike pruning). For instance, when compared to Distil-BERT, MUX-BERT ($N = 2$) does 1 point worse on QNLI and 2 points better on QQP while being equally fast and not requiring any additional unlabeled data.

More broadly, methods like CoFi, AutoTiny-BERT, and MobileBERT show that combining qualitatively different efficiency paradigms is a promising approach towards efficient models. For example, CoFi combines structured pruning and knowledge distillation, and AutoTinyBERT combines knowledge distillation and neural architecture search. This places importance on discovering new efficiency paradigms that dovetail with existing ones, rather than making incremental changes to current techniques.

Given that MIMO architectures like data multiplexing propose a novel direction towards efficiency, we demonstrate its complementary nature by combining it with pruning. We perform structured pruning using CoFi (sparsity of 0.6) on our MUX-PLMs. The resulting pruned model is $2\times$ faster than the original MUX-PLM while being as performant. We believe that the best combination of efficiency techniques is largely hardware and use-case dependent and we hope that MIMO architectures evolve in tandem with other approaches.

## 5.3 Effect of varying model size

In this section, we show that **our multiplexing techniques work on a host of model sizes** and report results for MUX-BERT on three models sizes, SMALL, BASE, and LARGE for $N = 2$ (Table 3). We report results for other values of $N$ in the appendix. MUX-BERT's performance is

close to that of BERT for all model sizes while having a significantly better throughput (the gap is less than 0.7 points for TOKEN tasks and 2.9 points for GLUE for close to twice the throughput). Multiplexing works effectively on all model sizes, with the drops with respect to BERT being 1.6 and 1.7 points on GLUE for SMALL and LARGE respectively. MUX-BERT's throughput is always $\approx 2\times$ that of BERT, which shows that a spectrum of MUX-PLM model sizes can be multiplexed during pre-training with competitive performance and with significantly higher throughput.

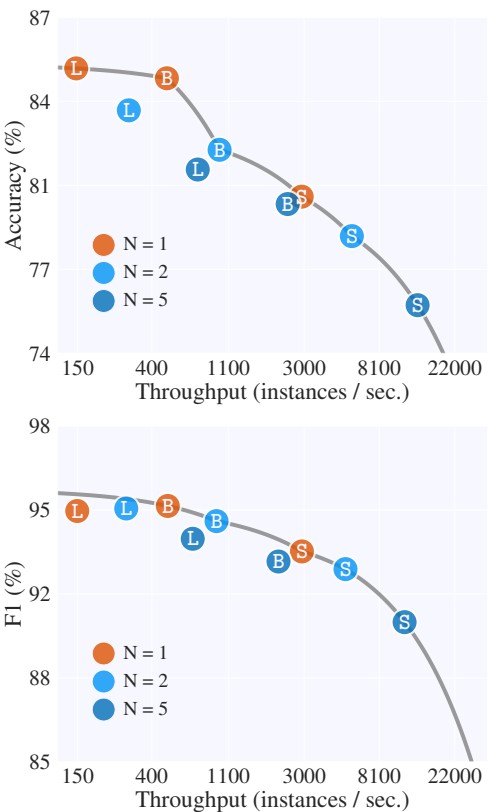

Figure 4: (Top) BERT GLUE performance and throughput and (Bottom) BERT Token task performance and throughput, for $N \in \{1, 2, 5, 10\}$ with the SMALL, BASE, and LARGE configurations (illustrated as S/B/L). All multiplexed models lie either on or very close to the Pareto frontier (shown in grey).

Pre-trained models typically have a performance-computational efficiency trade-off, with larger models having better performance but worse computational efficiency. MUX-PLMs offers a similar trade-off, with large $N$ leading to better throughput but lower performance. To understand this trade-off, we plot the performance and throughput of BERT and MUX-BERT for different model sizes and draw the pareto-optimal envelope (Figure 4). For any model on the envelope, no model

| Model | Mux ($N$) | MNLI | | | QQP | | |
|---|---|---|---|---|---|---|---|
| | | No Ens | Ens | Δ | No Ens | Ens | Δ |
| MUX-BERT | 2 | 80.6 | **81.2** | + 0.6 | 90.4 | **90.8** | + 0.4 |
| | 5 | 77.2 | **78.8** | + 1.6 | 88.8 | **89.7** | + 0.9 |
| | 10 | 73.6 | **74.8** | + 1.2 | 86.9 | **87.7** | + 0.8 |
| MUX-ELEC | 2 | 80.3 | **80.8** | + 0.5 | 90.6 | **90.9** | + 0.3 |
| | 5 | 77.0 | **78.4** | + 1.4 | 89.1 | **89.9** | + 0.8 |
| | 10 | 74.6 | **76.0** | + 1.4 | 87.6 | **88.3** | + 0.7 |

Table 4: Ensembling results for MUX-BERT and MUX-ELECTRA models for $N \in \{2, 5, 10\}$. *Ens* denotes Ensembling. Ensembling improves performance for all the models, with the gains increasing with increasing N. This suggests that the multiplexing approach can be naturally adapted to load-balancing applications, where the ensembling strategy can be changed based on demand.

has both better accuracy and throughput. Users would only choose models on the envelope because for every model within the envelope, there always exists a model on the envelope which has *both* better performance *and* throughput. We note that **all multiplexed models lie either on or very close to the Pareto frontier**, for both TOKEN and GLUE tasks. This suggests that given an accuracy threshold, MUX-PLM models will usually be faster than PLMs. For instance, if we wanted the highest throughput model with a performance $\geq 77\%$ on GLUE, the optimal BERT model is the SMALL configuration with a throughput of 2815 (in/s), but for the MUX-BERT model would be the $N = 2$ with the SMALL configuration, achieving a significantly higher throughput of 5539 (in/s).

## 5.4 Ensembling MUX-PLMs

As opposed to feeding $N$ different instances to MUX-PLMs to improve throughput, we consider an alternate setting where we feed the same instance $N$ times and build an ensemble by averaging the $N$ class logits to make a single prediction. We randomly permute the batch, after duplicating the instance $N$ times, to prevent distribution shift. We use the BASE size models for $N \in \{2, 5, 10\}$ for both MUX-BERT and MUX-ELECTRA (Table 4). **The ensemble model does significantly better than the non-ensemble variant** on both MNLI and QQP for all values of $N$ (e.g., 1.6 and 0.9 points on $N = 5$ MUX-BERT for the two tasks). We note that the improvement over the non-ensemble variant ($\Delta$) is better for higher $N$, due to the larger ensemble size. This result shows that non-ensemble variants are faster but perform slightly worse, while the ensemble variant performs better but is slower. A spectrum of models lie between

| Mux (N) | Model | Mux | Demux | GLUE | Token |
|---|---|---|---|---|---|
| 2 | MUX-BERT | Non-contextual | RSA-DeMUX | 82.5 | 95.2 |
| | Ablation 1 | Non-contextual | Prefix | **83.2** | **95.3** |
| | Ablation 2 | Contextual | RSA-DeMUX | 82.3 | 95.3 |
| 5 | MUX-BERT | Non-contextual | RSA-DeMUX | **80.3** | 93.6 |
| | Ablation 1 | Non-contextual | Prefix | 78.6 | 38.9 |
| | Ablation 2 | Contextual | RSA-DeMUX | 76.8 | **94.2** |
| 10 | MUX-BERT | Non-contextual | RSA-DeMUX | **77.8** | 91.6 |
| | Ablation 1 | Non-contextual | Prefix | 76.6 | 25.6 |
| | Ablation 2 | Contextual | RSA-DeMUX | 76.0 | **93.3** |

Table 5: Ablation analysis for MUX-BERT (base configuration) for $N \in \{2, 5, 10\}$. Across most configurations, the prefix demultiplexing variant performs worse than our proposed approach and fails to converge for token-level tasks for $N \in \{5, 10\}$ (underlined numbers). The new contextual multiplexing variant (Contextual) outperforms Non-contextual on token-level tasks.

these two extremes, where only a fraction of the $N$ multiplexed representations can be ensembled, allowing users to trade off performance and speed.

# 6 Analysis

## 6.1 Ablation study

We analyze multiplexing and demultiplexing components of MUX-PLMs and report the results in Table 5. We consider two variants, one which uses the prefix demultiplexing proposed in T-MUX instead of our proposed RSA-DeMUX and another which uses Contextual multiplexing instead of Non-contextual. We note that Variant 1, which uses prefix demultiplexing, performs worse than our MUX-BERT, other than for $N = 2$. In fact, Variant 1 does not converge for TOKEN tasks for $N = 5$ and $N = 10$ and performs 1.7 and 1.2 points worse on GLUE when compared to MUX-BERT.

Variant 2 uses Contextual multiplexing which takes into account other tokens present in the instance and also tokens present in the same position of other instances. This variant performs better than Non-contextual for TOKEN tasks (almost over 1.7 points on TOKEN for $N = 10$) but performs worse for GLUE tasks. We believe that Contextual multiplexing's better performance in TOKEN is because the model needs to make a prediction for every single position in the instance, which requires it to efficiently multiplex all token positions in the output. However, for GLUE tasks, the model needs to make a prediction only for the `[CLS]` token, for which Non-contextual multiplexing suffices.

| N | MUX-ELECTRA | | | MUX-BERT | | |
|---|---|---|---|---|---|---|
| | **Best ticket** | **Worst ticket** | $\Delta$ | **Best ticket** | **Worst ticket** | $\Delta$ |
| 2 | 83.1 | 82.0 | 1.1 | 83.4 | 81.8 | 1.6 |
| 5 | 80.5 | 78.9 | 1.6 | 80.9 | 79.7 | 1.2 |
| 10 | 79.0 | 77.3 | 1.7 | 78.8 | 77.0 | 1.8 |

Table 6: We consider 5 random seeds for every model variant, which can be viewed as lottery tickets as the seeds control the composition of N instances. We present the difference between the worst and the best-performing ticket across GLUE tasks and regularly see a $\geq 1$ point difference.

## 6.2 Effect of data sampling strategies during inference

During inference, our MUX-PLMs sample $N$ instances uniformly at random from the evaluation set. However, other data-sampling strategies such as clustering similar instances based on word-overlap could improve performance. We explore the effect of composition of $N$ instances on the performance of MUX-PLMs in Table 6. For each model variant, we consider 5 random seeds which can be viewed as lottery tickets (Frankle and Carbin, 2018). Since the random seed controls the composition of $N$ instances, we measure the difference ($\Delta$) between the best-performing ticket and the worst-performing ticket and average the performance for all the GLUE tasks. $\Delta$ is consistently greater than 1 point for all values of $N$ for both MUX-ELECTRA and MUX-BERT, and illustrates the importance of the composition of $N$ instances. An improved data sampling strategy could lead to improvements and we leave this to future work.

# 7 Conclusion

We introduce MUX-PLMs, a class of high-throughput pre-trained language models trained with data multiplexing, a multi-input multi-output (MIMO) architecture. Our MUX-PLMs models, trained with novel MIMO modules, are competitive with state-of-the-art PLMs on several downstream tasks while achieving a many-fold increase in inference throughput. MUX-PLMs, similar to standard PLMs, can be fine-tuned on any downstream task to yield high-throughput, high-performance models. We hope our work inspires future research in MIMO architectures for PLMs as a complementary efficiency paradigm to existing approaches.

## 8 Limitations

Our MUX-PLMs class of high-throughput high-performance pre-trained models demonstrates the efficacy of the MIMO paradigm for language models. However, MUX-PLMs need MIMO-style training for both pre-training and fine-tuning. It would be more efficient to introduce MIMO-style training *only* during the fine-tuning stage as it would allow stakeholders to rapidly create high-throughput models from any off-the-shelve pre-trained model. We conducted initial experiments in this setting but faced issues getting models to converge and perform well. This setting is incredibly challenging as off-the-shelf pre-trained models have been trained to only process one instance at a time and introducing our novel multiplexing and demultiplexing modules would create a large distribution shift for the model. We hope that future work focuses on innovative solutions to address these limitations.

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

| Hyperparameter | MUX-BERT | | | MUX-ELECTRA |
|---|---|---|---|---|
| | SMALL | BASE | LARGE | BASE |
| Number of layers | 4 | 12 | 24 | 12 |
| Hidden Size | 512 | 768 | 1024 | 768 |
| FFN intermediate hidden size | 2048 | 3072 | 4096 | 3072 |
| Attention heads | 8 | 12 | 16 | 12 |
| Attention head size | 64 | 64 | 64 | 64 |
| Mask percent | 15 | 15 | 15 | N/A |
| Learning Rate Decay | Linear | Linear | Linear | Linear |
| Warmup steps | 10000 | 10000 | 10000 | 10000 |
| Learning Rate | [1e-4, 5e-5] | [1e-4, 5e-5] | [1e-4, 5e-5] | [1e-4, 5e-5] |
| Adam $\epsilon$ | 1e-6 | 1e-6 | 1e-6 | 1e-6 |
| Adam $\beta_1$ | 0.9 | 0.9 | 0.9 | 0.9 |
| Adam $\beta_2$ | 0.999 | 0.999 | 0.999 | 0.999 |
| Attention Dropout | 0.1 | 0.1 | 0.1 | 0.1 |
| Dropout | 0.1 | 0.1 | 0.1 | 0.1 |
| Batch Size | 256 | 256 | 256 | 256 |
| Sequence Length | 512 | 512 | 512 | 512 |
| Train Steps | 1M | 1M | 1M | 1M |

Table 7: Pre-train hyper-parameters for MUX-BERT and MUX-ELECTRA models. We only report results for the Base configuration for MUX-ELECTRA models.

## A  Appendices

## B  Pre-training Details

We report all pre-training related hyper-parameters in Table 7. We primarily use the HuggingFace Transformers implementations for BERT and ELECTRA based models. All pre-training experiments were run on 8 A100 GPUs with distributed training. We run a small hyper-parameter search over over two learning rates. All pre-trained models are primed with the token retrieval task introduced in Murahari et al. (2022). We train on the Wikipedia and Bookscorpus datasets for up to 10000 training steps with a learning rate of $1e-4$, and with a sequence length of $512$.

For MUX-ELECTRA models, we don't train a generator as in the original ELECTRA work, but only use uniform-random token replacement. This is similar to what was used in ablations in ELECTRA (Clark et al., 2020). The generator randomly replaces $15\%$ of tokens in the input with other tokens in the vocabulary.

## C  Fine-tuning Details

We report all the fine-tuning related hyper-parameters in Table 8. We run a small hyper-parameter search on the learning rate, batch size and number of training steps for different tasks. All models were trained with half-precision. We report numbers on the validation split. For GLUE tasks, we use the default metrics in Wang et al. (2018) and use F1 for the token-level tasks. All fine-tuning experiments were trained on 1 V100 GPU.

**Speedup calculation** For all models, we calculate throughput (samples/second) on a single V100 GPU and report throughput gains with respect to the BERT$_{\text{BASE}}$ model. We calculate throughput by averaging across 3 different trials (1 trial = 200 mini-batches) and use a batch size of 128 and a sequence length of 128 following prior work (Xia et al., 2022). We measure throughput for sequence-classification tasks on QQP and measure throughput for token-level classification tasks on named entity recognition.

| Hyperparameter | Value |
|---|---|
| Learning Rate | [2e-5, 5e-5] |
| Adam $\epsilon$ | 1e-8 |
| Adam $\beta_1$ | 0.9 |
| Adam $\beta_2$ | 0.999 |
| Learning rate decay | Linear |
| Warmup fraction | 0.1 |
| Attention Dropout | 0.1 |
| Dropout | 0.1 |
| Weight Decay | 0 |
| Batch Size | [32, 128] for SMALL/ BASE, [16, 64] for LARGE |
| Train Steps | 2000 for RTE and WNLI |
| | 10000 for MRPC, COLA and STSB |
| | 20000 for NER, SST2, QNLI and POS |
| | [20000, 100000] for MNLI and QQP |
| Sequence Length | 128 |

Table 8: Fine-tune hyperparameters

## D Analysis details

### D.1 Ensembling results setup

We find that multiplexing the same instance by duplicating the instance N times leads to worse performance. This is likely because this input configuration is very out of distribution from what the multiplexed models are trained on. To address this, we randomly permute the instances in the batch after duplicating the instances N times. This ensures that the input to the multiplexer lies in a similar distribution to what the model was trained on.

### D.2 Muxology setup

To analyze the hidden states of pre-trained MUX-BERT models at different layers, we take the average absolute value of hidden states and every layer for both multiplexed and baseline models, across different configurations. To analyze the entropies of the attention distributions at different layers, we calculate the attention distribution across different attention heads for each position in the sequence length. To measure how peaky the attention distribution is likely to be, we calculate the entropies of the attention distributions at all positions and average across all the positions and across all the attention heads to get the average entropy for all layers. We conduct this analysis on WikiText-103 and average across all the samples in the evaluation split.

## E Task performance breakdown for all variants

| Model Size | N | MNLI | QQP | QNLI | MRPC | WNLI | STSB | RTE | SST2 | COLA | GLUE | GLUE$_{-\text{WNLI, COLA}}$ |
|---|---|---|---|---|---|---|---|---|---|---|---|---|
| SMALL | 1 | **77.86**$_{\pm0.0}$ | **88.99**$_{\pm0.0}$ | 84.00$_{\pm0.0}$ | 77.70$_{\pm0.0}$ | **56.34**$_{\pm0.0}$ | **84.25**$_{\pm0.0}$ | 62.45$_{\pm0.0}$ | **88.88**$_{\pm0.0}$ | 43.48$_{\pm0.0}$ | 73.77 | **80.59** |
| | 2 | 75.09$_{\pm0.1}$ | 88.88$_{\pm0.1}$ | **84.31**$_{\pm0.2}$ | 79.75$_{\pm0.7}$ | 50.99$_{\pm8.1}$ | 82.65$_{\pm0.3}$ | 55.52$_{\pm1.5}$ | 87.04$_{\pm0.7}$ | 30.64$_{\pm1.7}$ | 70.54 | 79.03 |
| | 5 | 70.50$_{\pm0.1}$ | 86.39$_{\pm0.1}$ | 81.23$_{\pm0.2}$ | 74.26$_{\pm1.0}$ | 54.65$_{\pm3.3}$ | 79.90$_{\pm0.2}$ | 58.56$_{\pm1.9}$ | 82.57$_{\pm0.3}$ | 12.78$_{\pm1.6}$ | 66.76 | 76.20 |
| | 10 | 61.98$_{\pm0.1}$ | 80.85$_{\pm0.1}$ | 63.47$_{\pm0.3}$ | 70.69$_{\pm0.9}$ | 56.62$_{\pm4.3}$ | 36.93$_{\pm1.0}$ | 53.57$_{\pm1.8}$ | 80.39$_{\pm0.4}$ | 1.10$_{\pm2.2}$ | 56.18 | 63.98 |
| BASE | 1 | 84.24$_{\pm0.0}$ | **91.19**$_{\pm0.0}$ | **90.54**$_{\pm0.0}$ | 87.75$_{\pm0.0}$ | 56.34$_{\pm0.0}$ | **89.18**$_{\pm0.0}$ | 63.18$_{\pm0.0}$ | **91.74**$_{\pm0.0}$ | **58.79**$_{\pm0.0}$ | 79.22 | **85.40** |
| | 2 | 80.59$_{\pm0.1}$ | 90.36$_{\pm0.1}$ | 88.17$_{\pm0.1}$ | 83.77$_{\pm1.4}$ | 50.70$_{\pm7.0}$ | 85.84$_{\pm0.1}$ | 58.19$_{\pm1.6}$ | 90.62$_{\pm0.6}$ | 55.61$_{\pm1.6}$ | 75.98 | 82.51 |
| | 5 | 77.18$_{\pm0.2}$ | 88.79$_{\pm0.1}$ | 85.58$_{\pm0.1}$ | 80.10$_{\pm0.6}$ | 53.52$_{\pm2.5}$ | 84.28$_{\pm0.2}$ | 59.13$_{\pm1.2}$ | 86.88$_{\pm0.4}$ | 12.33$_{\pm2.4}$ | 69.75 | 80.28 |
| | 10 | 73.62$_{\pm0.3}$ | 86.94$_{\pm0.1}$ | 82.08$_{\pm0.3}$ | 78.63$_{\pm0.6}$ | 52.68$_{\pm6.0}$ | 81.62$_{\pm0.2}$ | 58.27$_{\pm2.4}$ | 83.44$_{\pm0.6}$ | 0.00$_{\pm0.0}$ | 66.36 | 77.80 |
| LARGE | 1 | **85.79**$_{\pm0.0}$ | **91.46**$_{\pm0.0}$ | **92.29**$_{\pm0.0}$ | 83.82$_{\pm0.0}$ | 56.34$_{\pm0.0}$ | **89.53**$_{\pm0.0}$ | **66.06**$_{\pm0.0}$ | 91.40$_{\pm0.0}$ | 57.79$_{\pm0.0}$ | 79.39 | **85.76** |
| | 2 | 83.23$_{\pm0.2}$ | 90.85$_{\pm0.1}$ | 90.66$_{\pm0.2}$ | **84.90**$_{\pm0.8}$ | 56.34$_{\pm0.0}$ | 88.22$_{\pm0.2}$ | 59.21$_{\pm0.9}$ | 91.38$_{\pm0.4}$ | **57.89**$_{\pm1.5}$ | 78.08 | 84.06 |
| | 5 | 79.55$_{\pm0.2}$ | 89.37$_{\pm0.1}$ | 87.41$_{\pm0.2}$ | 83.77$_{\pm1.1}$ | 54.93$_{\pm0.0}$ | 85.86$_{\pm0.3}$ | 57.26$_{\pm2.0}$ | 88.65$_{\pm0.7}$ | 46.66$_{\pm0.9}$ | 74.83 | 81.70 |
| | 10 | 35.45$_{\pm0.0}$ | 63.18$_{\pm0.0}$ | 50.54$_{\pm0.0}$ | 68.38$_{\pm0.0}$ | **56.90**$_{\pm5.2}$ | 82.81$_{\pm0.2}$ | 52.13$_{\pm1.9}$ | 50.92$_{\pm0.0}$ | 1.87$_{\pm4.6}$ | 51.35 | 57.63 |

Table 9: We show the full GLUE results for MUX-BERT. We report the mean accuracy and standard deviation over 5 seeds. Extrema and values within their standard deviation are emphasized for each model size.

| Model Size | N | MNLI | QQP | QNLI | MRPC | WNLI | STSB | RTE | SST2 | COLA | GLUE | GLUE$_{-\text{WNLI, COLA}}$ |
|---|---|---|---|---|---|---|---|---|---|---|---|---|
| SMALL | 1 | **77.86** | **88.99** | 84.00 | 77.70 | 56.34 | **84.25** | 62.45 | **88.88** | 43.48 | **73.77** | **80.59** |
| | 2 | 75.21 | **89.01** | 84.61 | **80.64** | 61.97 | 82.97 | 58.12 | 87.84 | 33.08 | 72.61 | 79.77 |
| | 5 | 70.66 | 86.46 | 81.60 | 75.74 | 61.97 | 80.24 | 60.65 | 83.49 | 15.57 | 68.49 | 76.98 |
| | 10 | 62.17 | 80.93 | 63.85 | 71.81 | **63.38** | 38.20 | 55.96 | 80.96 | 2.63 | 57.77 | 64.84 |
| BASE | 1 | **84.24** | **91.19** | **90.54** | **87.75** | 56.34 | **89.18** | 63.18 | **91.74** | 58.79 | **79.22** | **85.40** |
| | 2 | 80.82 | 90.47 | 88.28 | 86.03 | **66.20** | 86.06 | 60.65 | 91.51 | 56.93 | 78.55 | 83.40 |
| | 5 | 77.66 | 88.89 | 85.70 | 81.13 | 59.15 | 84.47 | 60.65 | 87.50 | 15.79 | 71.22 | 80.86 |
| | 10 | 74.04 | 87.03 | 82.45 | 79.41 | 63.38 | 81.89 | 62.45 | 84.29 | 0.00 | 68.33 | 78.79 |
| LARGE | 1 | **85.79** | **91.46** | **92.29** | 83.82 | 56.34 | **89.53** | **66.06** | 91.40 | 57.79 | **79.39** | **85.76** |
| | 2 | 83.40 | 90.94 | 90.96 | **86.27** | 56.34 | 88.50 | 60.29 | **91.86** | **60.50** | 78.78 | 84.60 |
| | 5 | 79.69 | 89.43 | 87.81 | 84.80 | 57.75 | 86.49 | 60.65 | 89.45 | 47.56 | 75.96 | 82.62 |
| | 10 | 35.46 | 63.18 | 50.89 | 68.38 | **61.97** | 83.04 | 55.60 | 50.92 | 7.55 | 53.00 | 58.21 |

Table 10: We show the full GLUE results for MUX-BERT. We report the *maximum* accuracy over 5 seeds. Extrema are emphasized.

| N | MNLI | QQP | QNLI | MRPC | WNLI | STSB | RTE | SST2 | COLA | GLUE | GLUE$_{-\text{WNLI, COLA}}$ |
|---|---|---|---|---|---|---|---|---|---|---|---|
| 1 | **81.49**$_{\pm0.0}$ | **90.73**$_{\pm0.0}$ | **89.73**$_{\pm0.0}$ | 75.98$_{\pm0.0}$ | **56.34**$_{\pm0.0}$ | **87.73**$_{\pm0.0}$ | 57.76$_{\pm0.0}$ | **91.51**$_{\pm0.0}$ | **56.79**$_{\pm0.0}$ | **76.45** | 82.13 |
| 2 | 80.29$_{\pm0.2}$ | 90.58$_{\pm0.1}$ | 88.39$_{\pm0.2}$ | **83.73**$_{\pm0.7}$ | **57.18**$_{\pm2.1}$ | 86.80$_{\pm0.1}$ | **58.77**$_{\pm1.1}$ | 88.65$_{\pm0.4}$ | 51.92$_{\pm1.7}$ | 76.26 | **82.46** |
| 5 | 76.99$_{\pm0.2}$ | 89.08$_{\pm0.0}$ | 85.40$_{\pm0.3}$ | 80.25$_{\pm1.6}$ | **56.90**$_{\pm4.5}$ | 84.27$_{\pm0.2}$ | 57.26$_{\pm1.0}$ | 85.09$_{\pm1.0}$ | 26.89$_{\pm1.2}$ | 71.35 | 79.76 |
| 10 | 74.62$_{\pm0.2}$ | 87.63$_{\pm0.1}$ | 82.70$_{\pm0.2}$ | 77.89$_{\pm0.7}$ | 50.99$_{\pm4.9}$ | 81.96$_{\pm0.5}$ | **59.86**$_{\pm2.1}$ | 82.71$_{\pm0.5}$ | 27.76$_{\pm2.3}$ | 69.57 | 78.20 |

Table 11: We show the full GLUE results for MUX-ELECTRA$_{\text{BASE}}$. We report the mean accuracy and standard deviation over 5 seeds. Extrema and values within their standard deviation are emphasized for each model size.

| N | Retreival Rate | MNLI | QQP | QNLI | MRPC | WNLI | STSB | RTE | SST2 | COLA | GLUE | GLUE$_{-\text{WNLI, COLA}}$ |
|---|---|---|---|---|---|---|---|---|---|---|---|---|
| 2 | 0.0 | 83.23$_{\pm0.2}$ | 90.85$_{\pm0.1}$ | **90.66**$_{\pm0.2}$ | 84.90$_{\pm0.8}$ | **56.34**$_{\pm0.0}$ | 88.22$_{\pm0.2}$ | 59.21$_{\pm0.9}$ | **91.38**$_{\pm0.4}$ | 57.89$_{\pm1.5}$ | **78.08** | **84.06** |
| | 0.1 | **83.55**$_{\pm0.3}$ | 90.90$_{\pm0.1}$ | 90.58$_{\pm0.2}$ | **85.49**$_{\pm1.1}$ | **56.34**$_{\pm0.0}$ | 88.28$_{\pm0.2}$ | 57.76$_{\pm1.4}$ | 90.69$_{\pm0.8}$ | 59.36$_{\pm1.4}$ | **78.11** | 83.89 |
| | 0.2 | 83.50$_{\pm0.1}$ | **90.96**$_{\pm0.1}$ | 90.69$_{\pm0.2}$ | **84.95**$_{\pm0.5}$ | **56.34**$_{\pm0.0}$ | 88.28$_{\pm0.2}$ | 58.34$_{\pm1.6}$ | 90.69$_{\pm0.5}$ | 59.17$_{\pm1.5}$ | **78.10** | 83.92 |
| | 0.5 | 83.41$_{\pm0.2}$ | 90.91$_{\pm0.0}$ | 90.47$_{\pm0.1}$ | 85.25$_{\pm0.5}$ | **56.34**$_{\pm0.0}$ | 88.02$_{\pm0.1}$ | 59.35$_{\pm1.6}$ | 89.52$_{\pm0.6}$ | **59.41**$_{\pm2.0}$ | **78.08** | 83.85 |
| 5 | 0.0 | **79.55**$_{\pm0.2}$ | 89.37$_{\pm0.1}$ | **87.41**$_{\pm0.2}$ | 83.77$_{\pm1.1}$ | 54.93$_{\pm0.0}$ | 85.86$_{\pm0.3}$ | 57.26$_{\pm2.0}$ | 88.65$_{\pm0.7}$ | 46.66$_{\pm0.9}$ | **74.83** | **81.70** |
| | 0.1 | 79.49$_{\pm0.1}$ | 89.34$_{\pm0.1}$ | 87.25$_{\pm0.3}$ | 81.81$_{\pm1.3}$ | 53.24$_{\pm1.6}$ | 85.80$_{\pm0.2}$ | 55.60$_{\pm2.4}$ | 88.19$_{\pm0.7}$ | 47.60$_{\pm1.0}$ | 74.26 | 81.07 |
| | 0.2 | 79.37$_{\pm0.1}$ | **89.42**$_{\pm0.1}$ | 87.23$_{\pm0.3}$ | 82.40$_{\pm1.1}$ | 54.93$_{\pm0.0}$ | 85.85$_{\pm0.2}$ | 55.38$_{\pm2.6}$ | 87.84$_{\pm0.8}$ | 43.58$_{\pm1.2}$ | 74.00 | 81.07 |
| | 0.5 | 79.24$_{\pm0.1}$ | 89.30$_{\pm0.1}$ | 87.21$_{\pm0.3}$ | 82.06$_{\pm1.7}$ | **56.34**$_{\pm0.0}$ | **85.97**$_{\pm0.2}$ | 52.27$_{\pm4.0}$ | **88.58**$_{\pm0.6}$ | 47.01$_{\pm2.3}$ | 74.22 | 80.66 |
| 10 | 0.0 | **35.45**$_{\pm0.0}$ | **63.18**$_{\pm0.0}$ | 50.54$_{\pm0.0}$ | 68.38$_{\pm0.0}$ | **56.90**$_{\pm5.2}$ | 82.81$_{\pm0.2}$ | 52.13$_{\pm1.9}$ | 50.92$_{\pm0.0}$ | 1.87$_{\pm4.6}$ | 51.35 | 57.63 |
| | 0.1 | **35.45**$_{\pm0.0}$ | **63.18**$_{\pm0.0}$ | 50.65$_{\pm0.2}$ | 68.38$_{\pm0.0}$ | 54.93$_{\pm5.0}$ | 4.45$_{\pm1.5}$ | 51.48$_{\pm2.4}$ | 50.92$_{\pm0.0}$ | 1.34$_{\pm1.8}$ | 42.31 | 46.36 |
| | 0.2 | **35.45**$_{\pm0.0}$ | **63.18**$_{\pm0.0}$ | 50.21$_{\pm0.5}$ | 68.43$_{\pm0.8}$ | 54.65$_{\pm4.2}$ | 0.23$_{\pm1.5}$ | 52.35$_{\pm2.0}$ | **51.72**$_{\pm0.4}$ | 0.29$_{\pm2.7}$ | 41.83 | 45.94 |
| | 0.5 | **35.45**$_{\pm0.0}$ | **63.18**$_{\pm0.0}$ | 50.43$_{\pm0.4}$ | 68.38$_{\pm0.0}$ | 56.06$_{\pm0.6}$ | 82.01$_{\pm0.6}$ | 52.71$_{\pm0.0}$ | 50.92$_{\pm0.0}$ | 1.51$_{\pm1.7}$ | 51.18 | **57.58** |

Table 12: GLUE results for MUX-BERT$_{\text{LARGE}}$ when using a retrieval auxiliary objective during MLM pretraining with different trade-off rates to the MLM objective. We report the average accuracy over 5 seeds. Extrema and values within their standard deviation are emphasized for each value of N.

| N | Mux Strategy | MNLI | QQP | QNLI | MRPC | WNLI | STSB | RTE | SST2 | COLA | GLUE | GLUE$_{-\text{WNLI, COLA}}$ |
|---|---|---|---|---|---|---|---|---|---|---|---|---|
| | MUX-BERT | 80.59$_{\pm0.1}$ | 90.36$_{\pm0.1}$ | 88.17$_{\pm0.1}$ | 83.77$_{\pm1.4}$ | 50.70$_{\pm7.0}$ | 85.84$_{\pm0.1}$ | 58.19$_{\pm1.6}$ | 90.62$_{\pm0.6}$ | **55.61**$_{\pm1.6}$ | 75.98 | 82.51 |
| 2 | DataMUX | **81.64**$_{\pm0.2}$ | **90.67**$_{\pm0.1}$ | 88.39$_{\pm0.2}$ | **84.17**$_{\pm0.4}$ | 56.34$_{\pm0.0}$ | **86.36**$_{\pm0.2}$ | **60.87**$_{\pm0.7}$ | 90.50$_{\pm0.4}$ | 53.74$_{\pm1.0}$ | **76.96** | **83.23** |
| | Attention | 81.32$_{\pm0.2}$ | **90.65**$_{\pm0.0}$ | **88.77**$_{\pm0.1}$ | 80.88$_{\pm0.6}$ | 56.34$_{\pm0.0}$ | 86.25$_{\pm0.1}$ | 56.90$_{\pm1.2}$ | **91.06**$_{\pm0.2}$ | 47.15$_{\pm1.1}$ | 75.48 | 82.26 |
| | MUX-BERT | **77.18**$_{\pm0.2}$ | 88.79$_{\pm0.1}$ | **85.58**$_{\pm0.1}$ | **80.10**$_{\pm0.6}$ | 53.52$_{\pm2.5}$ | **84.28**$_{\pm0.2}$ | **59.13**$_{\pm1.2}$ | **86.88**$_{\pm0.4}$ | 12.33$_{\pm2.4}$ | 69.75 | **80.28** |
| 5 | DataMUX | 76.32$_{\pm0.1}$ | **89.13**$_{\pm0.1}$ | 84.22$_{\pm0.3}$ | 78.38$_{\pm0.9}$ | **59.44**$_{\pm3.5}$ | 81.78$_{\pm0.4}$ | 54.15$_{\pm1.3}$ | 86.17$_{\pm0.4}$ | 28.32$_{\pm0.8}$ | **70.88** | 78.59 |
| | Attention | **77.16**$_{\pm0.1}$ | 88.71$_{\pm0.0}$ | 84.33$_{\pm0.1}$ | 70.49$_{\pm0.6}$ | 54.08$_{\pm3.2}$ | 80.37$_{\pm0.3}$ | 54.44$_{\pm2.5}$ | 81.95$_{\pm0.3}$ | **34.67**$_{\pm1.2}$ | 69.58 | 76.78 |
| | MUX-BERT | **73.62**$_{\pm0.3}$ | 86.94$_{\pm0.1}$ | 82.08$_{\pm0.3}$ | **78.63**$_{\pm0.6}$ | 52.68$_{\pm6.0}$ | 81.62$_{\pm0.2}$ | **58.27**$_{\pm2.4}$ | **83.44**$_{\pm0.6}$ | 0.00$_{\pm0.0}$ | 66.36 | **77.80** |
| 10 | DataMUX | 72.74$_{\pm0.1}$ | 87.88$_{\pm0.1}$ | **82.28**$_{\pm0.2}$ | 77.30$_{\pm0.5}$ | 56.34$_{\pm0.0}$ | 78.07$_{\pm0.4}$ | 55.31$_{\pm1.2}$ | 82.36$_{\pm0.3}$ | 13.56$_{\pm3.0}$ | 67.32 | 76.56 |
| | Attention | 71.83$_{\pm0.2}$ | **88.00**$_{\pm0.0}$ | 81.46$_{\pm0.2}$ | 73.53$_{\pm0.5}$ | 53.24$_{\pm5.4}$ | **82.95**$_{\pm0.2}$ | 52.71$_{\pm0.0}$ | 81.28$_{\pm0.4}$ | **32.84**$_{\pm0.6}$ | **68.65** | 75.97 |

Table 13: GLUE results for MUX-BERT$_{\text{BASE}}$ using alternative multiplexing-demultiplexing strategies. We report the average accuracy over 5 seeds. Extrema and values within their standard deviation are emphasized for each value of N.

| Model Size | N | MNLI | QQP | QNLI | MRPC | WNLI | STSB | RTE | SST2 | COLA | GLUE | GLUE$_{-\text{WNLI, COLA}}$ |
|---|---|---|---|---|---|---|---|---|---|---|---|---|
| | 2 | **61.48**$_{\pm0.2}$ | **80.33**$_{\pm0.0}$ | **60.05**$_{\pm0.2}$ | 68.43$_{\pm0.5}$ | 56.34$_{\pm0.0}$ | **15.02**$_{\pm0.4}$ | 51.12$_{\pm0.6}$ | **79.75**$_{\pm0.3}$ | **8.22**$_{\pm0.7}$ | 53.42 | **59.45** |
| SMALL | 5 | 58.35$_{\pm0.2}$ | 77.50$_{\pm0.1}$ | 57.17$_{\pm0.3}$ | 68.38$_{\pm0.0}$ | 56.34$_{\pm0.0}$ | 11.31$_{\pm0.3}$ | 51.70$_{\pm1.3}$ | 77.78$_{\pm0.3}$ | 6.02$_{\pm0.7}$ | 51.62 | 57.46 |
| | 10 | 53.63$_{\pm0.2}$ | 77.03$_{\pm0.1}$ | 51.22$_{\pm0.3}$ | 68.38$_{\pm0.0}$ | **57.46**$_{\pm6.3}$ | 12.40$_{\pm1.3}$ | **52.35**$_{\pm2.7}$ | 50.92$_{\pm0.0}$ | 0.00$_{\pm0.0}$ | 47.04 | 52.28 |
| | 2 | **63.29**$_{\pm0.3}$ | **81.42**$_{\pm0.1}$ | **60.35**$_{\pm0.4}$ | 68.38$_{\pm0.2}$ | 56.90$_{\pm5.8}$ | **17.65**$_{\pm1.0}$ | 51.19$_{\pm1.7}$ | 80.78$_{\pm0.5}$ | 9.62$_{\pm1.5}$ | **54.40** | **60.44** |
| BASE | 5 | 60.67$_{\pm0.2}$ | 79.42$_{\pm0.1}$ | 59.77$_{\pm0.2}$ | **69.61**$_{\pm0.8}$ | 53.80$_{\pm7.3}$ | 14.92$_{\pm1.8}$ | 52.71$_{\pm0.8}$ | **81.15**$_{\pm0.6}$ | **10.35**$_{\pm1.7}$ | 53.60 | 59.75 |
| | 10 | 59.07$_{\pm0.2}$ | 78.22$_{\pm0.1}$ | 57.99$_{\pm0.5}$ | 68.38$_{\pm0.0}$ | **60.28**$_{\pm3.0}$ | 11.83$_{\pm0.6}$ | **53.07**$_{\pm1.1}$ | 78.35$_{\pm1.1}$ | 7.40$_{\pm1.7}$ | 52.73 | 58.13 |
| | 2 | **64.64**$_{\pm0.2}$ | **82.10**$_{\pm0.1}$ | **60.21**$_{\pm0.2}$ | **69.95**$_{\pm0.9}$ | 56.34$_{\pm0.0}$ | **21.62**$_{\pm0.4}$ | 52.71$_{\pm0.0}$ | **80.34**$_{\pm0.9}$ | 8.72$_{\pm2.1}$ | **55.18** | **61.65** |
| LARGE | 5 | 60.78$_{\pm0.3}$ | 78.56$_{\pm0.1}$ | **60.19**$_{\pm0.3}$ | 69.51$_{\pm0.5}$ | 56.34$_{\pm0.0}$ | 17.33$_{\pm1.1}$ | 52.71$_{\pm0.0}$ | 78.28$_{\pm0.8}$ | **10.63**$_{\pm2.7}$ | 53.81 | 59.62 |
| | 10 | 48.79$_{\pm0.6}$ | 68.41$_{\pm0.1}$ | 55.76$_{\pm0.8}$ | 68.58$_{\pm0.6}$ | **58.59**$_{\pm3.3}$ | 8.38$_{\pm1.1}$ | **54.95**$_{\pm0.9}$ | 64.82$_{\pm1.0}$ | 3.48$_{\pm3.9}$ | 47.97 | 52.81 |

Table 14: GLUE results for T-MUX with the original training recipe and implementation from Murahari et al. (2022). We report the average accuracy and standard deviation over 5 seeds. Extrema and values within their standard deviation are emphasized for each model size.