# OpenReview forum: "MUX-PLMs: Data Multiplexing for High-throughput Language Models"
_EMNLP/2023/Conference — EMNLP 2023 Findings_

### Official Review · Reviewer_29YX · 2023-08-01

**Soundness:** 3

**Excitement:**

4: Strong: This paper deepens the understanding of some phenomenon or lowers the barriers to an existing research direction.

**Paper Topic And Main Contributions:**

The paper proposes a novel pre-trained language model called MUX-PLMs, designed for multi-input multi-output scenarios. This method introduces a multiplexing layer at the input layer to compress multiple data streams and a demultiplexing layer at the output layer to decompress the hidden vectors. The experimental results demonstrate that MUX-PLMs, under both BERT and ELECTRA architectures, achieve comparable downstream performance to the original PLMs while significantly reducing the computational overhead.

**Questions For The Authors:**

1. Why is Table 2 not showing the performance of the Block Pruning, CoFi, and MobileBERT methods on the QQP task?
2. Section 5.4 demonstrates the performance of MUX-PLMs in an ensemble setting, where the computational overhead remains consistent with the original PLM. How does the performance of the ensemble MUX-PLMs compare to the original PLM?
3. In Table 5, when using the prefix as the demultiplexing layer with N = 5 or 10, the model fails to converge on the token classification task. Could you please provide an explanation for this phenomenon?

**Reasons To Accept:**

1. The authors present an interesting multi-input multi-output algorithm for PLMs, effectively reducing the computational costs of large language models.
2. Experimental results show that MUX-BERT and MUX-ELECTRA achieve significant reductions in computational costs while maintaining comparable performance to the original models, resulting in improved throughput.

**Reasons To Reject:**

1. The authors only validate the model's performance on GLUE, NER, and POS tagging tasks. Including more challenging tasks (e.g., SQuAD for question-answering) would strengthen the experimental results' persuasiveness.
2. According to the results in Table 5, the proposed contextual multiplex layer fails to effectively enhance the model's performance on downstream tasks.
3. The model's pre-training process involves two steps: token retrieval training followed by traditional language model objectives. Conducting an ablation study for both pre-training steps would make the experimental results more promising.
4. In the multi-input multi-output setting, different data in one batch can influence each other. Therefore, the authors should analyze and discuss the impact between different data within a batch.

**Reproducibility:**

4: Could mostly reproduce the results, but there may be some variation because of sample variance or minor variations in their interpretation of the protocol or method.

**Reviewer Confidence:**

3: Pretty sure, but there's a chance I missed something. Although I have a good feel for this area in general, I did not carefully check the paper's details, e.g., the math, experimental design, or novelty.

---

> ### Author Rebuttal · Authors · 2023-08-25
>
> We thank you for your valuable comments and we thank your support in bringing the MIMO paradigm to our community.
>
>
> > In Table 5, when using the prefix as the demultiplexing layer with N = 5 or 10, the model fails to converge on the token classification task. Could you please provide an explanation for this phenomenon?
>
> This is a great question. The prefix demultiplexer uses the output of the prefix tokens as conditioning in the demultiplexer. This makes the conditioning sensitive to the input and also causes stability issues when training the model across three different stages with three different objectives. This problem does not arise in the RSA-demux module as the N independent keys in the module transfer very well between the three different stages.
>
>  The token classification task requires the model to make predictions for every single position in the instance, which consequently requires it to multiplex and demultiplex all token positions in the output efficiently. This problem is magnified for higher values of N (5 or 10). Therefore, the prefix-based demultiplexer struggles to effectively demultiplex in the harder setting.
>
> > According to the results in Table 5, the proposed contextual multiplex layer fails to effectively enhance the model's performance on downstream tasks.
>
> We introduce the Contextual MUX module not only to improve performance but also to understand how design choices in the multiplexing module affect performance. Our experiments indicate that cleverly aggregating context does improve performance for token-level tasks, which require the model to make predictions for every single position in the instance, which consequently requires it to multiplex all token positions in the output efficiently. This might not be as critical for sequence classification tasks which require a single prediction for an entire sequence.
>
> > Section 5.4 demonstrates the performance of MUX-PLMs in an ensemble setting, where the computational overhead remains consistent with the original PLM. How does the performance of the ensemble MUX-PLMs compare to the original PLM?
>
> Ensembling MUX-PLMs improves performance over single MUX-PLMs but it is still lower compared to the original PLM.
>
> This is likely because a good ensemble involves a diverse set of functions that are accurate on different parts of the input space. Here, the function being ensembled is the model output for a particular index of the set of multiplexing functions. Therefore, we can think of it as an ensemble of N functions,  $f_1, \cdots f_N$, where $f_i$ is the output of MUX-BERT for the $i^{th}$ instance. These functions would be incentivized to be similar given the training objectives and this consequently would constitute a weaker ensemble.
>
> Importantly, we introduced the ensemble primarily to showcase that one could dynamically interpolate between accuracy and throughput by controlling which and how many instances are ensembled for a prediction. We discuss this further at the end of section 5.4.
>
> > Why is Table 2 not showing the performance of the Block Pruning, CoFi, and MobileBERT methods on the QQP task?
>
> We were not able to find official numbers for QQP for the validation split for these baselines.
>
> > The model's pre-training process involves two steps: token retrieval training followed by traditional language model objectives. Conducting an ablation study for both pre-training steps would make the experimental results more promising.
>
> We did run experiments without the token retrieval training but we could not get the model to converge. We therefore decided to not include that setting.

---

### Official Review · Reviewer_R2qS · 2023-08-04

**Soundness:** 3

**Excitement:**

3: Ambivalent: It has merits (e.g., it reports state-of-the-art results, the idea is nice), but there are key weaknesses (e.g., it describes incremental work), and it can significantly benefit from another round of revision. However, I won't object to accepting it if my co-reviewers champion it.

**Paper Topic And Main Contributions:**

This paper presents a method to augment language models’ throughputs using data multiplexing/demultiplexing. Like signals multiplexing in signal processing, authors’ method consists in superimposing entries, feed them to the model and disentangle the output of the model to recover each entry’s output. This way, the model can process multiple sentences at the same time. Method is not novel as it was introduced in a previous paper and even tested with Bert models (TMUX). Authors also propose a new framework to multiplex the signal, called contextual multiplexer, and claim their overall framework gets its inspiration from the RSA protocol, as it uses keys (vectors) to encode and decode signals. On paper their method seems to outperform the regular TMUX (the multiplexing framework that was previously published) but performs poorly compared to SOTA model compression methods (knowledge distillation). Their method requires a 3-stage-training: 1) autoencoder setting, to learn multiplex/demultiplex inputs/outputs. 2) Regular pretraining (with self-supervised objective). 3) Finetuning on downstream tasks. To my understanding, their method only differs from the TMUX by the length of the demultiplexing key.

**Questions For The Authors:**

1)	When using the contextual multiplexer, this should add some complexity overhead on the top of total inference time, and this is not considered in your speedup results. Correct?
2)	What is the Demux architecture? Like one in the TMUX paper?
3)	Are Trans_ins and Trans_ctx full transformers? Encoder-based transformers ? Decoder-based ?
4)	If I understand well, you claim that your method does not involve architectural change during the finetuning step. So do most of the baselines you listed: for example, it is true mobile-bert has a modified architecture, but during finetuning, it is finetuned like a regular bert model. Not sure I understand what you mean.
5)	Just an observation: I would not call your method RSA-like because RSA works in a very different way…

**Reasons To Accept:**

1)	Authors extend previous work done on data multiplexing by proposing a framework that outperforms its predecessor.
2)	Their contextual multiplexer seems to add some gains to the model’s final performance.
3)	Method can be applied to different architectures (BERT, ELECTRA).
4)	Method can be combined with other model compression methods.
5)	Paper written in a good English.

**Reasons To Reject:**

1)	There are some ambiguities in the paper: check “questions for authors” section.
2)	I don’t think there are many changes compared to the original TMUX version, a part from the contextual multiplexer (which is barely used by authors in the experiments) and the reduction of N de-multiplexing keys to one. Please correct me if I am wrong.
3)	Authors method performs worse than SOTA model compression methods such as MobileBERT and TinyBERT: while the former only increases models’ speed inference, the latter also reduce models’ sizes. On a size, we have a method that accelerates models’ inference and on the other, we have methods that compress models’ size and accelerate them even more. Which one would you pick?
4)	I think this framework could have been tested on decoder-based transformers as well. I don’t see any of that on the paper.
5)	From the ablation study, it seems like this method is only effective (compared to the original TMUX) when used on token-level tasks.

**Reproducibility:**

4: Could mostly reproduce the results, but there may be some variation because of sample variance or minor variations in their interpretation of the protocol or method.

**Reviewer Confidence:**

4: Quite sure. I tried to check the important points carefully. It's unlikely, though conceivable, that I missed something that should affect my ratings.

---

> ### Author Rebuttal · Authors · 2023-08-25
>
> Thank you for your detailed comments. We really appreciate it.
> We would like to clarify a few misunderstandings:
>
>
> >  Method is not novel as it was introduced in a previous paper and even tested with Bert models (TMUX).
>
>
> MUX-PLMs, to the best of our knowledge, is the **first pre-trained MIMO-style (Multi-input Multi-output) class of models**. Pre-training MIMO-style models is not trivial as it requires significant changes to both the pre-trained objective and the pre-training procedure to ensure training stability and competitive performance.
>
>
> **The difference between T-MUX and MUX-PLMs is akin to the difference between the Transformer and BERT paper, which were both novel and impactful contributions in their own right.** T-MUX has a transformer backbone but is not pre-trained and **Murahari. et al. do not propose either pre-training strategies or pre-trained models.** T-MUX models perform very poorly compared to MUX-PLMs and other SOTA models.
>
>
> The goal of our paper is to show that MIMO-style architectures can achieve competitive performance compared to SOTA models and to that end, we explored different design choices such as the demultiplexing and multiplexing modules and the pre-training recipe/ algorithm.
>
>
> >  What is the Demux architecture? Like one in the TMUX paper?
>
>
> Our novel RSA-demux demultiplexing module (Figure 2, Section 3.2) enables large-scale pre-training of MUX-PLMs and is significantly faster and more performant than the prefix-based demultiplexer introduced by T-MUX (Section 3.1).
>
>
> Please note that there are several major differences between the two demultiplexing modules. Firstly, the prefix-based demultiplexer adds a set of prefix tokens (Section 3.1) and uses the output of the prefix tokens as conditioning for the demultiplexer to disentangle the representations. Our novel RSA-demux module instead learns a set of N demultiplexing vectors (Figure 2, Section 3.2) along with the other layers in the module to disentangle the multiplexed representations. This implies the following:
>
> a) The conditioning for the demultiplexing module does not vary for every input unlike the prefix-based demultiplexer and this leads to better training stability. This is because the N keys transfer well across the three different training stages. For example, the ablations in Table 5 indicate that the prefix-based demultiplexer from T-MUX does not even converge for token-level tasks and performs worse than our RSA-demux in almost all settings.
>
> b) This demultiplexing module is significantly faster as observed in the speedup column in Table 1.
>
>
> **Differences between T-MUX and MUX-PLMs**
>
> > From the ablation study, it seems like this method is only effective (compared to the original TMUX) when used on token-level tasks.
>
> Our method consists of all the three components listed below, and each component is in itself novel and performant. Therefore, the ablation in Table 2 does not suggest that our method only does well on token-level tasks but suggests that using the contextual multiplexer together with the RSA-demultiplexer module improves performance on token-level tasks.
>
>
> 1. RSA-demux module:  Our novel RSA-demux demultiplexing module enables large-scale pre-training of MUX-PLMs and is significantly faster and more performant than prior work.
>
> 2. Pre-training MIMO style models: We have addressed this in our response above. T-MUX is not a pre-trained model and also does not provide any pre-training objectives or strategies. MUX-PLMs represent the first pre-trained MIMO style models competitive with SOTA models.
>
> 3. Contextual multiplexer:  We introduce our novel contextual multiplexer module to explore the importance of various design choices for effective multiplexing. Our contextual multiplexer module highlights the importance of aggregating context for token-level tasks but does not improve performance for sequence classification tasks.
>
>
> > Authors method performs worse than SOTA model compression methods such as MobileBERT and TinyBERT: while the former only increases models’ speed inference, the latter also reduce models’ sizes. On a size, we have a method that accelerates models’ inference and on the other, we have methods that compress models’ size and accelerate them even more. Which one would you pick?
>
>
> As noted by all the reviewers, MUX-PLMs and MIMO architectures more generally are complementary to existing model compression methods.
>
> We demonstrate this in Table 2 by combining MUX-BERT (N = 2) with CoFi, a SOTA structured pruning approach. The resulting model is 2x faster than the MUX-BERT model and 4x faster than the baseline, while being equally performant, and demonstrates that the **acceleration ratios for both methods multiply without a significant drop in performance**.
>
> Therefore, future work could easily extend MUX-PLMs to yield MUX-TinyBERT or MUX-MobileBERT to get the best of both paradigms i.e. highly compressed models with very high throughput.
>
>
> > I think this framework could have been tested on decoder-based transformers as well. I don’t see any of that on the paper.
>
> Thanks for the suggestion. We agree that the framework could be extended to decoder-based transformers. Given the compute budget constraints in an academic setting, we focussed our attention on thoroughly exploring MIMO-style pre-trained models for transformer-encoder models but we expect future work to trivially extend MUX-PLMs to decoder-based transformers.
>
> > Are Trans_ins and Trans_ctx full transformers? Encoder-based transformers ? Decoder-based ?
>
> > When using the contextual multiplexer, this should add some complexity overhead on the top of total inference time, and this is not considered in your speedup results. Correct?
>
>
> Trans_ins and Trans_ctx are **single** transformer **layers**. Therefore, the contextual multiplexer module in total has two transformer layers.
>
>
> Adding the two additional layers does increase overhead but not significantly.  The overhead increases with increasing N as the two transformer layers have to process a very large batch size for a large N.
>
>
> We include the speedup numbers for Table 5 below:
>
>
> | MUX(N) | Model      | MUX            | DEMUX     | Speedup |
> |--------|------------|----------------|-----------|---------|
> | 2      | MUX-BERT   | Non-contextual | RSA-demux | 2       |
> | 2      | Ablation 1 | Non-contextual | Prefix    | 1.9     |
> | 2      | Ablation 2 | Contextual     | RSA-demux | 1.7     |
> | 5      | MUX-BERT   | Non-contextual | RSA-demux | 4.9     |
> | 5      | Ablation 1 | Non-contextual | Prefix    | 4.4     |
> | 5      | Ablation 2 | Contextual     | RSA-demux | 4.1     |
> | 10     | MUX-BERT   | Non-contextual | RSA-demux | 9.8     |
> | 10     | Ablation 1 | Non-contextual | Prefix    | 7.9     |
> | 10     | Ablation 2 | Contextual     | RSA-demux | 5.9     |
>
> > Just an observation: I would not call your method RSA-like because RSA works in a very different way…
>
> We were inspired by RSA when coming up with the RSA-demux demultiplexing module. If the reviewers find it confusing, we will be happy to change the name.

---

### Official Review · Reviewer_dyvA · 2023-08-05

**Soundness:** 3

**Excitement:**

3: Ambivalent: It has merits (e.g., it reports state-of-the-art results, the idea is nice), but there are key weaknesses (e.g., it describes incremental work), and it can significantly benefit from another round of revision. However, I won't object to accepting it if my co-reviewers champion it.

**Paper Topic And Main Contributions:**

The paper explores the application of Data Multiplexing technology in pre-trained language models to enhance model throughput. The authors propose a three-stage MUX-PLM training method, along with the design of novel MUX and DeMUX modules. Experimental validation demonstrates that MUX-PLM can effectively boost model throughput with minimal impact on performance. Furthermore, this technique can be integrated with existing compression methods to achieve superior results.

**Questions For The Authors:**

- What is the primary role of the Hadamard product in the MUX module? Can it be understood as a type of identifier for each sample, similar to how position encoding is used in the transformer?
- The pre-training of MUX-PLM involves two stages. Does this require a longer training period compared to the standard PLM training? Could you provide a comparison of the training computational times?
- Are the poor results on WNLI and CoLA due solely to the small size of the data sets, or is it because MUX-PLM is not suitable for these types of tasks? Could you sample a subset from data sets like SST-2 to confirm whether MUX-PLM performs poorly on all small data sets?
- What is the model size of T-MUX in Table 1? Is it the same size as MUX-BERT and MUX-ELECTRA?
- How is the DeMUX function calculated in Line 288?
- Section 5.4 analyzes the results of ensemble learning. Although the performance has improved compared to without ensemble learning, it is still not as good as the BERT model with equivalent computational cost. What could be the reason for this?


**Reasons To Accept:**

- The paper is well-written and easy to read.
- Research in the field of MIMO holds significant potential and could serve as an important supplement to existing acceleration methods.
- The experiments have demonstrated that the impact on performance is minimal when throughput is increased.

**Reasons To Reject:**

- Some of the contributions made by the paper are not fully substantiated by the experiments. The article introduces the Contextual MUX module, but it does not perform as well as the original MUX module on the GLUE benchmark. Furthermore, the Contextual MUX was not employed in the main experiment.
- Compared to other acceleration methods, MUX-PLM does not demonstrate strong competitiveness in terms of performance. It may not yet be adopted by users from a practical perspective, indicating room for further optimization. Additionally, MUX-PLM may result in higher training costs. Achieving different acceleration ratios requires **pre-training** different MUX-PLMs, while existing compression methods can obtain models with different acceleration ratios through **fine-tuning** a certain PLM.

**Reproducibility:**

3: Could reproduce the results with some difficulty. The settings of parameters are underspecified or subjectively determined; the training/evaluation data are not widely available.

**Reviewer Confidence:**

4: Quite sure. I tried to check the important points carefully. It's unlikely, though conceivable, that I missed something that should affect my ratings.

---

> ### Author Rebuttal · Authors · 2023-08-25
>
> We thank you for your insightful comments and we thank your support in bringing the MIMO paradigm to our community.
> > Additionally, MUX-PLM may result in higher training costs.
>
> > The pre-training of MUX-PLM involves two stages. Does this require a longer training period compared to the standard PLM training? Could you provide a comparison of the training computational times?
>
> **Both the baseline PLMs and MUX-PLMs are pre-trained with the same number of optimization steps** (Please refer to Table 7 for other details). We also use the same hyper-parameter sweep for fine-tuning MUX-PLMs and PLMs (Please refer to Table 8).
>
>
> The first stage (“Token Retrieval” in Figure 1) is trained for 10K optimization steps. We pre-train all models for 1M optimization steps. The first stage is a couple of orders of magnitude cheaper than pre-training and hardly leads to any overhead. Appendix B contains more details.
>
>
> > Achieving different acceleration ratios requires pre-training different MUX-PLMs while existing compression methods can obtain models with different acceleration ratios through fine-tuning a certain PLM.
>
>
> As noted by all the reviewers, MUX-PLMs and MIMO architectures more generally are complementary to existing model compression methods.
>
>
> We demonstrate this in Table 2 by combining MUX-BERT (N = 2) with CoFi, a SOTA structured pruning approach. The resulting model is 2x faster than the MUX-BERT model and 4x faster than the baseline while being equally performant. This demonstrates that the **acceleration ratios for both methods multiply without a significant drop in performance**.
>
>
> We believe MIMO architectures will continue to evolve in tandem with the current and future efficiency paradigms to leverage the multiplication of these acceleration ratios to get faster and more performant models.
>
>
> Finally, MUX-PLMs are the first pre-trained MIMO style models in the community and we believe the community will make training of MUX-PLMs more efficient, where potentially existing PLMs are quickly adapted to yield MUX-PLMs.
>
> > The article introduces the Contextual MUX module, but it does not perform as well as the original MUX module on the GLUE benchmark.
>
> We introduce the Contextual MUX module not only to improve performance but also to understand how design choices in the multiplexing module affect performance. Our experiments indicate that cleverly aggregating context does improve performance for token-level tasks, which require the model to make predictions for every single position in the instance, which consequently requires it to multiplex all token positions in the output efficiently. This might not be as critical for sequence classification tasks which require a single prediction for an entire sequence.
>
>
> We will improve the writing to reflect this.
>
>
> > What is the primary role of the Hadamard product in the MUX module? Can it be understood as a type of identifier for each sample, similar to how position encoding is used in the transformer?
>
>
> Your understanding is very accurate, it is indeed a type of identifier for each sample that helps us generate a compressed and multiplexed representation for the set of N-independent samples.
>
>
> > Are the poor results on WNLI and CoLA due solely to the small size of the data sets, or is it because MUX-PLM is not suitable for these types of tasks? Could you sample a subset from data sets like SST-2 to confirm whether MUX-PLM performs poorly on all small data sets?
>
>
> We would like to clarify that the performance on WNLI and CoLA are *not poor but instead have high variance*.
>
>
> Tables 9 - 14 in Appendix E contain performance metrics and error bars for each individual task for various model configurations. Consider the WNLI column in Table 9, Our MUX-PLM models perform very well compared to the non-multiplexed baseline. However, the variance across the 5 seeds is very high.
>
>
> > What is the model size of T-MUX in Table 1? Is it the same size as MUX-BERT and MUX-ELECTRA?
>
>
> Yes, it is the exact same size as MUX-BERT and MUX-ELECTRA. We have listed the size and the other hyper-parameter details in Table 7.
>
>
> > Section 5.4 analyzes the results of ensemble learning. Although the performance has improved compared to without ensemble learning, it is still not as good as the BERT model with equivalent computational cost. What could be the reason for this?
>
>
> A good ensemble involves a diverse set of functions that are accurate on different parts of the input space. Here, the function being ensembled is the model output for a particular index of the set of multiplexing functions. Therefore, we can think of it as an ensemble of N functions,  f_1, .. f_N, where f_i is the output of MUX-BERT for the ith instance. These functions would be incentivized to be similar given the training objectives and this consequently would constitute a weaker ensemble.
>
>
> We introduced the ensemble primarily to showcase that one could dynamically interpolate between accuracy and throughput by controlling which and how many instances are ensembled for a prediction. We discuss this further at the end of section 5.4.
>
>
> > How is the DeMUX function calculated in Line 288?
>
>
> The DEMUX function is instantiated with a standard feedforward network containing linear layers, layer norm layers, gelu activation, and some skip connections. We will improve the writing to clarify this.

---

### Meta-Review · Senior_Area_Chairs · 2023-09-30

**Recommendation:** 3

**Metareview:**

The paper introduces MUX-PLMs, a pre-trained language model designed to enhance throughput in multi-input multi-output scenarios. The approach uses data multiplexing to process multiple inputs simultaneously, promising improved inference speeds. Experiments focus primarily on the GLUE benchmark, with the paper arguing for both the novelty and effectiveness of MUX-PLMs. Three reviewers have provided their assessments.

Strengths:

1. Novel Approach: The authors' introduction of a MUX and DeMUX architecture in the realm of PLMs is noteworthy. This innovation aims to address the growing demand and computational cost challenges associated with large language models.
2. Positive Results: Experiments indicate that MUX-PLMs, particularly under BERT and ELECTRA architectures, can maintain performance levels comparable to traditional PLMs while offering significant reductions in computational overhead.
3. Clarity: The paper is well-structured, making the content easily accessible to readers. Several reviewers highlighted its readability.
4. Broad Applicability: The proposed method is shown to work across different architectures like BERT and ELECTRA and can also be combined with other compression methods.

Weaknesses:

1. Insufficient Validation: The model's performance validation seems restricted, with emphasis on GLUE, NER, and POS tagging tasks. An exploration into more diverse and challenging tasks would have strengthened the study's credibility.
2. Performance of Contextual MUX: The paper introduces the Contextual MUX module, but it reportedly does not outperform the original MUX module. Its absence in primary experiments and lackluster results question its utility.
3. Comparison with SOTA: The method's performance, when juxtaposed with state-of-the-art model compression techniques (like MobileBERT and TinyBERT), is reportedly not competitive. Reviewers questioned the practical adoption of MUX-PLMs given these findings.
4. Ambiguities & Lack of Depth: Several concerns were raised regarding ambiguities in the paper and a perceived lack of depth in specific sections. For instance, the method's uniqueness compared to the earlier TMUX version was questioned. A more profound exploration of the impact of different data within a batch in a multi-input setting would also have been beneficial.
5. Training Cost & Flexibility: The potential increased training costs associated with MUX-PLMs and the need to pre-train different MUX-PLMs for different acceleration ratios could be a limitation compared to existing methods that fine-tune a single PLM.

The paper, with its focus on MUX-PLMs, offers a unique lens into improving language model throughput via data multiplexing. While the premise is exciting and there are promising results, there are notable gaps in validation, comparative analysis, and practical implications that should be addressed. Future revisions should consider expanding the experimental range, addressing ambiguities, and elucidating the tangible benefits of adopting MUX-PLMs over existing methods.

---

### Meta-Review · Area_Chair_itce · 2023-10-05

**Recommendation:** 3

**Metareview:**

This paper presents MUX-PLMs, a pre-trained language model strategically crafted to optimize throughput in multi-input multi-output scenarios. By harnessing data multiplexing, this approach concurrently handles multiple inputs, offering the prospect of significantly enhanced inference speeds. The experimental evaluation primarily centers around the GLUE benchmark, demonstrating the novelty and efficacy of MUX-PLMs. The insights of three reviewers further enrich the paper's assessment.

---

### Decision · Program_Chairs · 2023-10-07

**Decision:**

Accept-Findings

**Comment:**

The paper introduces MUX-PLMs, a pre-trained language model designed to enhance throughput in multi-input multi-output scenarios. The approach uses data multiplexing to process multiple inputs simultaneously, promising improved inference speeds. Experiments focus primarily on the GLUE benchmark, with the paper arguing for both the novelty and effectiveness of MUX-PLMs. Three reviewers have provided their assessments.

Strengths:

1. Novel Approach: The authors' introduction of a MUX and DeMUX architecture in the realm of PLMs is noteworthy. This innovation aims to address the growing demand and computational cost challenges associated with large language models.
2. Positive Results: Experiments indicate that MUX-PLMs, particularly under BERT and ELECTRA architectures, can maintain performance levels comparable to traditional PLMs while offering significant reductions in computational overhead.
3. Clarity: The paper is well-structured, making the content easily accessible to readers. Several reviewers highlighted its readability.
4. Broad Applicability: The proposed method is shown to work across different architectures like BERT and ELECTRA and can also be combined with other compression methods.

Weaknesses:

1. Insufficient Validation: The model's performance validation seems restricted, with emphasis on GLUE, NER, and POS tagging tasks. An exploration into more diverse and challenging tasks would have strengthened the study's credibility.
2. Performance of Contextual MUX: The paper introduces the Contextual MUX module, but it reportedly does not outperform the original MUX module. Its absence in primary experiments and lackluster results question its utility.
3. Comparison with SOTA: The method's performance, when juxtaposed with state-of-the-art model compression techniques (like MobileBERT and TinyBERT), is reportedly not competitive. Reviewers questioned the practical adoption of MUX-PLMs given these findings.
4. Ambiguities & Lack of Depth: Several concerns were raised regarding ambiguities in the paper and a perceived lack of depth in specific sections. For instance, the method's uniqueness compared to the earlier TMUX version was questioned. A more profound exploration of the impact of different data within a batch in a multi-input setting would also have been beneficial.
5. Training Cost & Flexibility: The potential increased training costs associated with MUX-PLMs and the need to pre-train different MUX-PLMs for different acceleration ratios could be a limitation compared to existing methods that fine-tune a single PLM.

The paper, with its focus on MUX-PLMs, offers a unique lens into improving language model throughput via data multiplexing. While the premise is exciting and there are promising results, there are notable gaps in validation, comparative analysis, and practical implications that should be addressed. Future revisions should consider expanding the experimental range, addressing ambiguities, and elucidating the tangible benefits of adopting MUX-PLMs over existing methods.|This paper presents MUX-PLMs, a pre-trained language model strategically crafted to optimize throughput in multi-input multi-output scenarios. By harnessing data multiplexing, this approach concurrently handles multiple inputs, offering the prospect of significantly enhanced inference speeds. The experimental evaluation primarily centers around the GLUE benchmark, demonstrating the novelty and efficacy of MUX-PLMs. The insights of three reviewers further enrich the paper's assessment.